# Radar and Jammer Intelligent Game under Jamming Power Dynamic Allocation

**Jie Geng** [1] , **Bo Jiu** [1,*], **Kang Li** [1], **Yu Zhao** [1], **Hongwei Liu** [1] and **Hailin Li** [2]

1   National Laboratory of Radar Signal Processing, Xidian University, Xi'an 710071, China
2   Beijing Institute of Tracking and Telecommunication Technology, Beijing 100094, China
*   Correspondence: bojiu@xidian.edu.cn

**Abstract:** In modern electronic warfare, the intelligence of the jammer greatly worsens the anti-jamming performance of traditional passive suppression methods. How to actively design anti-jamming strategies to deal with intelligent jammers is crucial to the radar system. In the existing research on radar anti-jamming strategies' design, the assumption of jammers is too ideal. To establish a model that is closer to real electronic warfare, this paper explores the intelligent game between a subpulse-level frequency-agile (FA) radar and a transmit/receive time-sharing jammer under jamming power dynamic allocation. Firstly, the discrete allocation model of jamming power is established, and the multiple-round sequential interaction between the radar and the jammer is described based on an extensive-form game. A detection probability calculation method based on the signal-to-interference-pulse-noise ratio (SINR) accumulation gain criterion (SAGC) is proposed to evaluate the game results. Secondly, considering that the competition between the radar and the jammer has the feature of imperfect information, we utilized neural fictitious self-play (NFSP), an end-to-end deep reinforcement learning (DRL) algorithm, to find the Nash equilibrium (NE) of the game. Finally, the simulation results showed that the game between the radar and the jammer can converge to an approximate NE under the established model. The approximate NE strategies are better than the elementary strategies from the perspective of detection probability. In addition, comparing NFSP and the deep Q-network (DQN) illustrates the effectiveness of NFSP in solving the NE of imperfect information games.

**Keywords:** electronic warfare; intelligent game; jamming power dynamic allocation; neural fictitious self-play; deep reinforcement learning; Nash equilibrium

## 1. Introduction

In modern electronic warfare, the radar faces great challenges from different advanced jamming types [1]. Among different jamming types, main lobe jamming is especially difficult to deal with because the jammer and the target are close enough and both in the main lobe of the radar antenna [2].

Radar anti-main lobe jamming technologies mainly include passive suppression and active antagonism. The passive suppression methods mean that, after the radar is jammed, it can filter out the jamming signal by finding the separable domain between the target echo and the jamming signal [3–6]. In contrast to the passive suppression methods, active antagonism requires the radar to take measures in advance to avoid being jammed [7]. Common active countermeasures include frequency agility, waveform agility, pulse repetition frequency (PRF) agility, and joint agility [8]. Since the frequency-agile (FA) radar can randomly change the carrier frequency in each transmit pulse, it is difficult for the jammer to intercept and jam the radar, which is considered to be an effective means of anti-main lobe jamming [9,10]. In [11], frequency agility combined with the PRF jittering method for the radar transmit waveform was proposed to resist deception jamming. In [12], the

authors proposed a moving target detection algorithm under the background of deception jamming based on FA radar.

The key to FA radar anti-jamming is the frequency-hopping strategy. For the purposes of the electronic counter-countermeasures (ECCM) considered in this paper, the radar needs to take different frequency-agile strategies to deal with different jamming strategies. How to design frequency-agile strategies according to the jammer's actions is of vital importance. For an effective anti-jamming system, the information about the environment and the jammer must be known; otherwise, the judgment of the radar is not credible [13]. Therefore, some researchers have introduced reinforcement learning (RL) algorithms to design anti-jamming strategies for FA radar. In [14], the authors designed a novel frequency-hopping strategy for cognitive radar against the jammer, and the radar does not need to know the operating mode of the jammer. The signal-to-interference-pulse-noise ratio (SINR) as a reward function was used in [14], and the interaction between the radar and the jammer was achieved by two methods, Q-learning and the deep Q-network (DQN), to learn the attack strategy of the jammer to avoid the radar being jammed. In [15], the authors designed an anti-jamming strategy for FA radar against spot jamming based on the DQN approach. Unlike the SINR reward function adopted in [14], Reference [15] used the detection probability as a reward for the radar to learn the optimal anti-jamming strategy. In [16], a radar anti-jamming scheme with the joint agility of the carrier frequency and pulse width was proposed. Different from the anti-jamming strategy design for the pulse-level FA radar in [14,15], Reference [17] studied the anti-jamming strategy for the subpulse-level FA radar, where the carrier frequency of the transmit signal can be changed both within and between pulses. In addition, a policy-gradient-based RL algorithm known as proximal policy optimization (PPO) was adopted in [17] to further improve the anti-jamming performance of the radar.

Currently, most of the research assumes that the jamming strategy is static, which means that the jammer is a dumb jammer who adopts a fixed jamming strategy. However, the jammer can also adaptively learn jamming strategies according to the radar's actions [18,19]. How to model and study intelligent games between the radar and the jammer is of great significance to modern electronic warfare.

The game analysis framework can generally be used to model and deal with multi-agent RL problems [20]. It is feasible to apply game theory to model the relationship between the radar and the jammer. In [21], the competition between the radar with constant false alarm processing and the self-protection jammer was considered based on the static game, and the Nash equilibrium (NE) was studied for different jamming types. In [22], the competition was also modeled by the static game, and the NE strategies could be obtained. In [23,24], a co-located multiple-input multiple-output (MIMO) radar and a smart jammer were considered, and the competition was modeled based on the dynamic game. From the perspective of mutual information, the NE of the radar and the jammer were solved.

Although the jammer is considered as a player, which has the same intelligence level as the radar, the established model is too ideal in the above-mentioned work. For example, the work based on static games cannot characterize the sequence decision-making between the radar and the jammer, and the work based on dynamic games only considers a single-round interaction. In real electronic warfare, the competition between the radar and the jammer is a multiple-round interaction with imperfect information [25]. In addition, with the advancement of jamming technology, the jammer can transmit spot jamming, which aims at multiple frequencies simultaneously [26]. How to establish a more realistic electronic warfare model becomes a preliminary step for designing anti-jamming strategies for the radar.

Therefore, this paper considered a signal model of the jammer as transmitting spot jamming with its central frequency aiming at different frequencies simultaneously, and the jamming power of each frequency can be arbitrarily allocated under the constraint condition. Extensive-form games [27] are proposed to model the relationship of the multiple-round sequence decision-making between the radar and the jammer. Imperfect information was

also considered through the characteristics of the partial observation of two-player games. Under this model, the NE strategies of the competition between the radar and the jammer with jamming power dynamic allocation can be investigated. The main contributions of this work are summarized as follows:

- A mathematical model of jamming power discrete allocation is established. Different action spaces of the jammer can be obtained for different quantization steps of power. The smaller the quantization step, the larger the action space of the jammer. When the number of available actions is more, the jammer could find the optimal jamming strategy, and the conclusion is proven by simulation.
- A detection probability calculation method based on the SINR accumulation gain criterion (SAGC) is proposed. After the radar receives a target echo, it judges whether each subpulse is retained or discarded through the SAGC. The specific calculation procedure is that the radar uses the subpulse and the subpulse with the same carrier frequency retained in the past to calculate the coherent integration. If the SINR is improved, the subpulse is retained; otherwise, the subpulse is discarded. At the end of one coherent processing interval (CPI), the coherent integration results obtained from the retained subpulses are used to calculate the detection probability based on the SINR-weighting-based detection (SWD) [17,28] algorithm.
- Extensive simulations were carried out to demonstrate the competition results. Specifically, the training curves of the detection probability of the radar and whether the game between the radar and the jammer can converge to an NE under different quantization steps of power were investigated. The simulation results showed that: (1) the proposed SAGC outperformed another criterion; (2) the game can achieve an approximate NE; if the jammer action space is larger, the game can achieve an NE because the jammer can explore the best action; (3) the approximate NE strategies are better than elementary strategies from the perspective of detection performance.

The remainder of this paper is organized as follows. In Section 2, the signal model of the radar and the jammer is introduced and the jamming power allocation model is proposed. In Section 3, the game elements for the radar and the jammer are designed in detail. In Section 4, the deep reinforcement learning (DRL) and NFSP algorithms are described and the overall confrontation process between the radar and the jammer is given. Section 5 shows the results of the competition between the radar and the jammer under the system model, and Section 6 summarizes the work of this paper.

## 2. System Model

Consider a game between a subpulse-level FA radar [29] and a jammer. Compared with the pulse-level FA radar, the subpulse-level FA radar can further improve the anti-jamming performance of the radar [17].

### 2.1. The Signal Model of the Radar

Assume that the radar transmits $N$ pulses in one CPI, which contains $K$ subpulses. The mathematical expression of the $n$th pulse is

$$
\begin{aligned}
s_{TX}(t,n) &= \sum_{k=0}^{K-1} u(t - nT_r - kT_c) \exp[j2\pi(f_0 + a_{n,k}\Delta f)t] \\
&= \sum_{k=0}^{K-1} \mathrm{rect}(t - nT_r - kT_c) \exp\left[j\pi\gamma(t - nT_r - kT_c)^2\right] \exp[j2\pi(f_0 + a_{n,k}\Delta f)t],
\end{aligned}
\tag{1}
$$

where $u(t) = \mathrm{rect}(t)\exp\left[j\pi\gamma t^2\right]$ is the complex envelope of the signal, $T_r$ and $T_c$ are the pulse repetition interval (PRI) and the subpulse width, respectively, $f_0$ denotes the initial carrier frequency, and $\Delta f$ is the step size between two subcarriers. As for $a_{n,k}$, it represents the frequency hopping code; assume that the number of available frequencies

for the radar is $M$, then $a_{n,k} \in \{0, 1, \cdots, M-1\}$. Here, rect$(t)$ is a rectangular function and is described by

$$\text{rect}(t) = \left\{ \begin{array}{l} 1, 0 < t < T_c \\ 0, \text{otherwise} \end{array} \right. . \tag{2}$$

Take $K = 4$ and $M = 5$ as an example. The time–frequency diagram of the radar transmit waveform is illustrated in Figure 1.

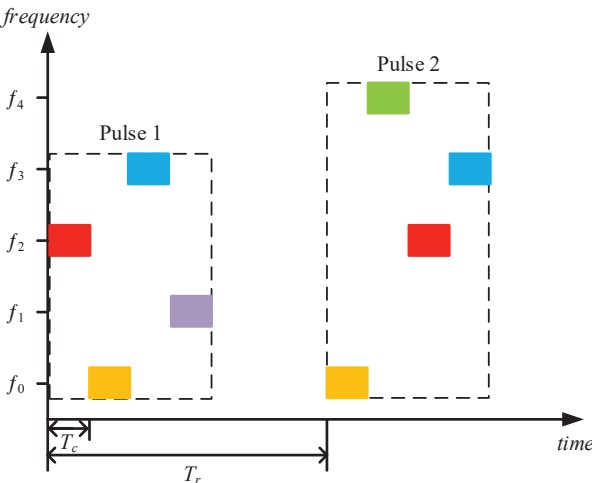

**Figure 1.** Time–frequency diagram of the subpulse FA radar waveform.

Assume that there is only one target. The $n$th target echo can be expressed as follows:

$$s_{RX}(t, n) = \sum_{k=0}^{K-1} \sigma_k u(t - nT_r - kT_c - \tau_0) \exp[j2\pi(f_0 + a_{n,k}\Delta f)(t - \tau_0)], \tag{3}$$

where $\sigma_k$ is the subpulse echo amplitude corresponding to the carrier frequency of that subpulse and $\tau_0 = 2R/c$ denotes the time delay of the target echo.

### 2.2. The Signal Model of the Jammer

The jammer considered in this paper is a self-protection jammer that works in a transmit/receive time-sharing mode. Therefore, the jammer cannot receive and transmit signals at the same time. The jamming type is spot jamming. To accurately implement spot jamming, the jammer needs to intercept a portion of the radar waveform and measure its carrier frequency, which is called look-through [30]. After that, the jammer transmits a jamming signal based on the carrier frequency of the intercepted radar waveform. Therefore, the signal model of the jammer consists of two parts: interception and transmission.

For the convenience of analysis, it is assumed that the interception duration $T_l$ of the jammer is an integer multiple of the radar subpulse duration $T_c$:

$$T_l = ZT_c \ (0 \leq Z \leq K). \tag{4}$$

Suppose that the delay of the $n$th pulse transmitted by the radar reaching the jammer is $\tau'$; the interception action of the jammer with respect to this pulse can be expressed as

$$J_{RX}(t, n) = \text{rect}\left[\frac{T_c(t - nT_r - \tau')}{T_l}\right] s_{TX}(t - \tau', n). \tag{5}$$

If the jammer transmits spot jamming that aims at $G$ frequencies simultaneously, the expression of the jamming signal is

$$J_{TX}(t) = \sum_{j=0}^{G-1} \xi_j \text{rect}\left(\frac{t - nT_r - \tau' - T_l}{KT_c - T_l} T_c\right) \exp(j2\pi f_j t), \tag{6}$$

where $f_j$ is the $j$th central frequency of the spot jamming and $\xi_j$ is the Gaussian process with variable variance representing the jamming power allocated to the $j$th frequency.

The time–frequency diagram of the signal transmitted by the jammer is shown in Figure 2. The dashed box indicates that the period is an interception. Different colors mean that the jammer transmits the spot jamming with its central frequency aiming at different frequencies simultaneously in the remaining subpulses.

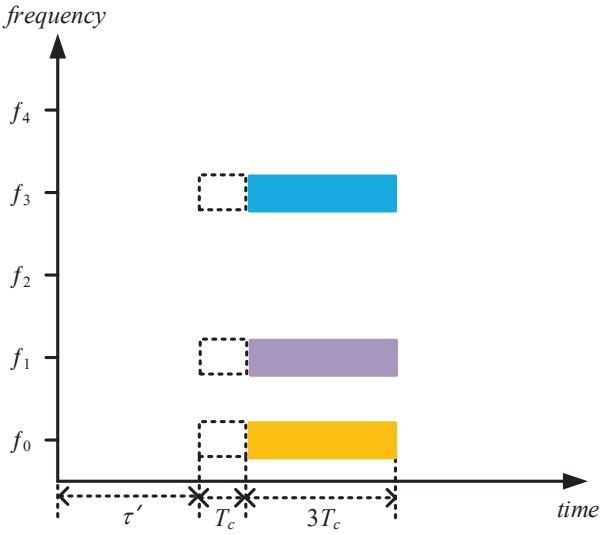

**Figure 2.** Time–frequency diagram of jammer's transmitted signal.

### 2.3. The Discrete Allocation Model of Jamming Power

As described in Section 2.2, the jammer transmits the jamming signal, which aims at multiple frequencies simultaneously and allocates the total power reasonably to these frequencies. To simplify the analysis, it was assumed that the total power of the jammer is normalized to 1. Besides, assume that the jammer cannot allocate its power to each frequency arbitrarily, which is restricted by $P_{\min}(0 < P_{\min} \leq 1)$. In other words, $P_{\min}$ is the smallest unit of power allocation. Therefore, the power allocated by the jammer for each frequency is an integer multiple of $P_{\min}$. The smallest unit $P_{\min}$ is defined as the "quantization step". According to the total power and "quantization step", the number of samples of jamming power is

$$N = \frac{1}{P_{\min}}. \tag{7}$$

The number of frequencies available for the radar is $M$; denote the number of power samples distributed by the jammer to these frequencies as $N_0, N_1, \cdots, N_{M-1}$, then the percentage of power allocated to each frequency is

$$P_i = N_i \times P_{\min}, i \in \{0, 1, \cdots, M-1\}. \tag{8}$$

The allocation model should satisfy the following constraints:

$$\text{s.t.} \begin{cases} 0 \leq N_i \leq N \\ \sum_{i=0}^{M-1} N_i = N \end{cases}. \tag{9}$$

The competition between the radar and the jammer is dynamic, which means they both optimize their strategies to maximize their performance. As for the jammer, its central frequency selection and power allocation strategy is not fixed and can be optimized by interacting with the radar.

## 3. Game Elements Design for Radar and Jammer

In a complex electronic warfare environment, the confrontation between the radar and the jammer is often multi-round and can be regarded as a sequential decision-making process. The interaction process between the radar and the jammer can be described as follows. The radar transmits the signal, and the jammer makes a decision based on the intercepted partial information of the radar. The radar analyses the behavior of the jammer or the possible jamming strategy based on the interfered echoes and improves the transmitting waveform in the next pulse to achieve the anti-jamming objective.

Each pulse transmitted by the radar corresponds to one competition between the radar and the jammer. At the end of one CPI, the radar will evaluate the anti-jamming performance of the entire process based on the information of all previous pulses. Extensive-form games are a model involving the sequential interaction of multiple agents [31], which can conveniently describe the relationship between the radar and the jammer. The essential elements of the game include actions, information states, and payoff functions.

Generally, the interaction process between the radar and the jammer can be modeled by game theory, in which the radar and the jammer are players in a game.

### 3.1. Radar Actions

The target of the subpulse-level FA radar is to adopt an appropriate frequency-hopping strategy to deal with interference, and each transmitted pulse is one competition, so the action of the radar is defined as the carrier frequency combination of subpulses. Given the number of subpulses $K$ in one pulse and the number of available carrier frequencies $M$, the action of the radar can be expressed as $\mathbf{a}_t^r = \left[ a_{t,1}^r, \cdots, a_{t,K}^r \right]$, which is a vector with size $1 \times K$. Each element $a_{t,i}^r \in \{0, \cdots, M-1\}$ represents the subcarrier of the $i$th subpulse of the $t$th pulse. For example, $a_{t,i}^r = 2$ indicates that the subcarrier is $f_2$. Based on the number of subpulses and the available frequencies, it can be known that the total number of actions of the radar is $\mathcal{A}_R = M^K$.

### 3.2. Jammer Actions

The action of the jammer consists of two parts: interception and transmission. To simplify the analysis, assume that the total duration of these two actions is equal to the duration of the radar pulse. According to the number of subpulses, the interception action of the jammer takes any value in set $\{0, 1, \cdots, K\}$, which denotes the number of look-through subpulses. If the value of the interception action is $K$, then the jammer does not transmit any jamming signal and only executes the look-through operation. The jammer transmits the jamming signal referring to the number of power samples allocated to different frequencies. Based on the number of available frequencies for the radar, $[N_0, \cdots, N_{M-1}]$ can represent this part of the action. The value of $N_i$ is related to the quantization step of the jamming power $P_{\min}$ and should satisfy the allocation model in Section 2.3. Combining the two actions of interception and transmission, the complete action of the jammer is a vector with size $1 \times (M + 1)$. It is worth noting that when the quantization step remains unchanged, unless the jammer intercepts all subpulses, the number of the jammer action in the second part is the same. Take $K = 2$, $M = 2$, $P_{\min} = 0.5$ as an example. According to the jamming power allocation model, it can be known that there are three allocation schemes, which is the number of the jammer actions in the second part, as shown in Table 1.

**Table 1.** Jamming power allocation schemes.

| index | 1 | 2 | 3 |
|---|---|---|---|
| scheme | [0, 2] | [2, 0] | [1, 1] |

The interception action can be 0, 1, and 2. Only when the interception code is 2, the second part of the jammer action is all 0. Under other codes, the transmission action can be any of the cases in Table 1. Therefore, the total number of jammer actions is $\mathcal{A}_J = 2 \times 3 + 1 = 7$. The complete actions of the jammer are shown in Table 2.

**Table 2.** The action set of the jammer.

| action number | 1 | 2 | 3 | 4 | 5 | 6 | 7 |
|---|---|---|---|---|---|---|---|
| action vector | [1, 0, 2] | [1, 2, 0] | [1, 1, 1] | [2, 0, 0] | [0, 0, 2] | [0, 2, 0] | [0, 1, 1] |

### 3.3. Information States

In the competition between the radar and the jammer, the radar decides the action at the next moment according to the behavior of the jammer, and so does the jammer. The information state is defined as the player's actions and partial observations of adversary actions at all historical times. Partial observation makes the player unable to fully obtain the opponent's actions, which reflects the imperfect information of the game. When calculating the information state of the jammer at time $t$, the radar has executed action $\mathbf{a}_t^r$. Since the action of the jammer always lags behind the radar in timing, the current radar action $\mathbf{a}_t^r$ is not available to the jammer. This also reflects the existence of imperfect information. The information states of the radar and the jammer are given as follows:

$$\mathbf{s}_t^r = \left[ \mathbf{a}_0^r, \mathbf{o}_0^j, \cdots, \mathbf{a}_{t-1}^r, \mathbf{o}_{t-1}^j \right], \tag{10}$$

$$\mathbf{s}_t^j = \left[ \mathbf{o}_0^r, \mathbf{a}_0^j, \cdots, \mathbf{o}_{t-1}^r, \mathbf{a}_{t-1}^j \right], \tag{11}$$

where $\mathbf{o}_{t-1}^j$ denotes the partial observation of the jammer action by the radar at time $t - 1$. $\mathbf{o}_{t-1}^r$ represents the partial observation of the radar action by the jammer at time $t - 1$.

### 3.4. Payoff Functions

The payoff function is used to evaluate the value of the agent's policy. After the agent makes an action according to the information state, it will obtain a feedback signal from the environment. The agent judges the value of that action according to the feedback information to guide subsequent learning. Therefore, the agent will formulate a payoff function as the feedback. Through the payoff function, it can achieve the expected objective. Detection probability is an important performance indicator of the radar, which can be used as the feedback for the anti-jamming strategies' design. However, in practical signal processing, the radar calculates the detection probability based on the information of all pulses after one CPI ends. The game between the radar and the jammer is based on a single pulse, so taking the detection probability as a payoff function will bring the problem of a sparse reward. For each echo received by the radar, the SINR of the echo can be calculated. The existence of jamming signals will reduce the SINR. Thus, it is feasible to use the SINR as a reward to guide anti-jamming strategies' learning for the radar and can avoid the sparse reward. The calculation formulas [32] for the signal power and jamming power of the $k$th subpulse echo are

$$P_{r_k} = \frac{P_T G_R^2 \lambda_k^2 \sigma_k}{(4\pi)^3 R^4}, \tag{12}$$

$$P_{j_k} = \frac{P_J G_R G_J \lambda_k^2}{(4\pi)^2 R^2}, \tag{13}$$

where $P_T$ and $G_R$ are the radar transmission power and antenna gain, respectively, $R$ represents the distance between the radar and the target, $\lambda_k$ and $\sigma_k$ are the wavelength and radar cross section (RCS) corresponding to the $k$th subpulse carrier frequency, and $P_J$ and $G_J$ are the jammer transmission power and antenna gain. Therefore, the mathematical expression for calculating the SINR of the $k$th subpulse is

$$\text{SINR}_k = \frac{P_{r_k}}{P_N + P_{j_k} \cdot \mathbf{1}(f_k = f_j)}, \tag{14}$$

where $P_{r_k}$ and $P_{j_k}$ are the signal power and jamming power of the $k$th subpulse echo, respectively; $P_N$ is the system noise power of the radar receiver, and it can be estimated by

$$P_N = \bar{k} T_0 B_n, \tag{15}$$

where $\bar{k} = 1.38 \times 10^{-23}$ J/K is the Boltzmann constant, $T_0 = 290$ K is the effective noise temperature, and $B_n$ is the bandwidth of a subpulse.

In (14), $P_{j_k}$ is the jamming power entering the radar receiver, but it exists only when the central frequency $f_j$ of the jamming signal is equal to the subpulse carrier frequency $f_k$. Otherwise, it is 0. Therefore, $\mathbf{1}(x)$ can be expressed by

$$\mathbf{1}(x) = \begin{cases} 1, \text{if } x \text{ is true} \\ 0, \text{elsewhere} \end{cases}. \tag{16}$$

Therefore, the payoff function of the radar can be expressed as follows:

$$R_t^r = \sum_{k=0}^{K-1} \text{SINR}_k. \tag{17}$$

Due to the hostile relationship between the radar and the jammer, they can be regarded as a two-player zero-sum (TPZS) game, so the payoff function of the jammer is given as follows:

$$R_t^j = -\sum_{k=0}^{K-1} \text{SINR}_k. \tag{18}$$

### 3.5. Detection Probability Calculation Method Based on SINR Accumulation Gain Criterion

In Section 3.4, the target echo power, jamming power, and noise power can be estimated. Based on this information, the coherent integration of each carrier frequency is obtained according to the SINR accumulation gain criterion (SAGC). Then, the detection probability is calculated by the SWD algorithm [17,28]. The calculation step of the SAGC is given below:

(1) Let $\text{SINR}_k^n$ denote the coherent integration of $f_k$ from $n$ pulses. Here, we take two carrier frequencies $f_1, f_2$, two subpulses, and one CPI containing four pulses as an example. Therefore, the value of $k$ is 1 and 2, and the value of $n$ is 1 to 4. Let the initial thresholds of the SINR of these two frequencies be $T_1$ and $T_2$, respectively.

(2) After the radar receives the first pulse echo, if the carrier frequencies of the two subpulses are $[f_2, f_1]$, the signal power is $[P_{r_2}, P_{r_1}]$, and the noise power is $[P_N, P_N]$, the jamming power of each subpulse is determined as $[P_{j_2}, P_{j_1}]$ based on the central frequency and power allocation schemes of the jamming signal. According to the above information of the first pulse echo, the coherent integration of each frequency can be calculated (since there is only one pulse and the carrier frequency is different, the SINR is calculated directly).

$$\text{SINR}_1^1 = \frac{P_{r_1}}{P_N + P_{j_1}}, \tag{19}$$

$$\text{SINR}_2^1 = \frac{P_{r_2}}{P_N + P_{j_2}}. \tag{20}$$

Judgment: if $\text{SINR}_1^1 > T_1$, retain the subpulse whose carrier frequency is $f_1$ and update the value of $T_1$ with $\text{SINR}_1^1$. Otherwise, discard the subpulse whose carrier frequency is $f_1$, and still use the initial $T_1$ as the threshold. In the same way, it is determined whether the subpulse whose carrier frequency is $f_2$ is reserved or discarded. Assume that both subpulses are retained here, then $T_1 = \text{SINR}_1^1$, $T_2 = \text{SINR}_2^1$.

(3) After the radar receives the second echo, if the frequency is $[f_1, f_2]$, the signal power is $[P_{r_1}, P_{r_2}]$, and the noise power is $[P_N, P_N]$, the jamming power of each subpulse is determined as $[P_{j_1}, P_{j_2}]$ according to the jamming signal. Each subpulse is coherently integrated with the same carrier frequency as the subpulse reserved in the first pulse.

Firstly, add the first subpulse to calculate the coherent integration of $f_1$:

$$\text{SINR}_1^2 = \frac{\left(\sqrt{P_{r_1}} + \sqrt{P_{r_1}}\right)^2}{P_N + P_{j_1} + P_N + P_{j_1}}, \tag{21}$$

and if $\text{SINR}_1^2 > T_1$, reserve the subpulse with carrier frequency $f_1$ in the second echo and update the value of $T_1$ with $\text{SINR}_1^2$. Otherwise, discard the subpulse, and do not update the value of $T_1$.

Next, append the second subpulse to compute the coherent integration of $f_2$:

$$\text{SINR}_2^2 = \frac{\left(\sqrt{P_{r_2}} + \sqrt{P_{r_2}}\right)^2}{P_N + P_{j_2} + P_N + P_{j_2}}, \tag{22}$$

and if $\text{SINR}_2^2 > T_2$, retain the subpulse with carrier frequency $f_2$ in the second echo and update the value of $T_2$ with $\text{SINR}_2^2$. Otherwise, discard the subpulse, and the value of $T_2$ is not updated.

(4) After receiving the third echo, the radar takes the same operation: adding subpulses in turn to calculate the coherent integration of each frequency and comparing with the thresholds to determine whether to retain the subpulses and update the thresholds. Until the end of one CPI, the obtained SINR is used as the final coherent integration of each frequency.

It is important to note that, although the symbols of the jamming powers of different echoes are the same, their values are different and depend on the specific jamming situation.

SAGC focuses on the impact of a single subpulse on the overall effect, rather than just the subpulse itself. Another advantage of SAGC is that the coherent integration of all frequencies is immediately available as the last pulse is judged.

## 4. Approximate Nash Equilibrium Solution Based on Neural Fictitious Self-Play

### 4.1. Deep Reinforcement Learning

RL problems can be described by a Markov decision process (MDP) [33]. At time $t$, the agent observes the environment state $s_t$ and selects action $a_t$ according to the strategy $\pi(a_t|s_t)$. After acting, the agent will obtain a reward signal $r_{t+1}$ indicating the quality of the action and make the environment enter a new state $s_{t+1}$. The objective of the agent is to maximize the cumulative reward through continuous interaction with the environment, which is given in (23):

$$\pi^* = \underset{\pi}{\arg\max}\, \mathbb{E}[R_t|\pi], \tag{23}$$

where $R_t = \sum_{k=0}^{\infty} \gamma^k r_{t+1+k}$ is a discounted long-term reward with $\gamma \in [0, 1)$ denoting the discount factor.

Value-based and policy-gradient-based methods are two commonly used methods for solving RL problems. The value-based methods need to estimate the state–action value function and then obtain the optimal strategy through the value function. The policy-gradient-based methods calculate the gradient of the objective function on the policy parameters to solve the optimal strategy. The policy-gradient-based methods are usually

used to deal with high-dimensional and continuous action space problems. Since the action space of the radar and jammer considered in this paper is discrete and not so large, the value-based method was used to solve the optimal strategy.

The long-term expected reward when starting in a specific state $s$ following the policy $\pi$ is called the state value function, which is defined as

$$V_\pi(s) = \mathbb{E}_\pi \left[ \sum_{k=0}^{\infty} \gamma^k r_{t+1+k} | s_t = s \right]. \tag{24}$$

The state–action value function denotes the long-term expected return after executing action $a$ in state $s$ according to policy $\pi$, which is defined as

$$Q_\pi(s, a) = \mathbb{E}_\pi \left[ \sum_{k=0}^{\infty} \gamma^k r_{t+1+k} | s_t = s, a_t = a \right]. \tag{25}$$

The relationship between the state value function and the state–action value function is

$$V_\pi(s) = \mathbb{E}_{a \sim \pi(a|s)} [Q_\pi(s, a)]. \tag{26}$$

$Q_\pi(s, a)$ can guide the agent's decision. If the agent adopts the greedy strategy, it chooses the action that maximizes $Q(s, a)$ at each moment. If the agent executes the $\varepsilon - \text{greedy}(Q)$ strategy, it selects the action that maximizes $Q(s, a)$ with probability $1 - \varepsilon$ and randomly chooses an action from the action space with probability $\varepsilon$. The agent follows the $\varepsilon - \text{greedy}(Q)$ policy to balance exploration and exploitation when it acts [33].

$$\varepsilon - \text{greedy}(Q) \leftarrow \begin{cases} \arg\max_{a'} Q(s, a'), \text{with probability } 1 - \varepsilon \\ \text{random}(\mathcal{A}), \quad \text{with probability } \varepsilon \end{cases} \tag{27}$$

Estimates for the optimal action values can be learned using Q-learning [34]. In standard Q-learning, the estimation accuracy is increased by visiting states during the exploration phase and replacing the value of each state–action pair using the Bellman optimality equation:

$$Q(s, a) \leftarrow Q(s, a) + \alpha \left[ r + \gamma \max_{a'} Q(s', a') - Q(s, a) \right], \tag{28}$$

where $\alpha \in [0, 1)$ is the learning rate.

Deep reinforcement learning (DRL) combines deep neural networks and RL, introduces an approximate representation of the value function, and solves the problem of instability in the learning process based on two key technologies of experience replay and target network [20]. DQN [35] is a typical value-based DRL algorithm, which means it needs to estimate the state–action value function from the samples. The loss function to update the parameters of the neural network of the DQN is given in (29):

$$\mathcal{L}\left(\theta^Q\right) = \mathbb{E}_{\{s,a,r,s'\} \in \mathcal{D}_{RL}} \left\{ \left[ r + \max_{a'} Q\left(s', a' \middle| \theta^{Q'}\right) - Q\left(s, a \middle| \theta^Q\right) \right]^2 \right\}. \tag{29}$$

*4.2. Neural Fictitious Self-Play*

The confrontation between the radar and the jammer has the characteristics of multiple-round sequential decision-making, which allows us to model their interactions with extensive-form games. Moreover, due to the transmit/receive time-sharing working mode of the jammer, the game has imperfect information. NFSP is an end-to-end DRL algorithm for solving the approximate NE of extensive-form games with imperfect information and does not need any prior knowledge [36]. NFSP includes DRL and supervised learning (SL) when solving strategies, and both of them can only be applied to problems with a

discrete action space. Combined with the model established in this paper, NFSP is feasible to solve NE.

NFSP agents learn directly from the experience of interacting with other agents in the game based on DRL. Each NFSP agent contains a memory buffer $\mathcal{D}_{RL}$ that stores the transition experience $\{s_t, a_t, r_{t+1}, s_{t+1}\}$ and a memory buffer $\mathcal{D}_{SL}$ that stores the best response $\{s_t, a_t\}$. NFSP treats these buffers as two separate datasets suitable for deep reinforcement learning and supervised classification, respectively. The agent trains the value network parameters $\theta^Q$ from the data in $\mathcal{D}_{RL}$ using an off-policy RL algorithm based on experience replay. The value network defines the agent's best response policy $\varepsilon - \text{greedy}(Q)$. The agent trains a separate neural network to imitate its own past best response behavior using supervised classification data in $\mathcal{D}_{SL}$. This network achieves the mapping of states to action probabilities. Define the action probability distribution of the network output as the agent's historical average policy $\Pi$. Based on the above two strategies, the NFSP agent chooses action $a_t$ in state $s_t$ from a mixture of its two policies, and it can be expressed as follows:

$$\sigma \leftarrow \begin{cases} \varepsilon - \text{greedy}(Q), \text{with probability } \eta \\ \Pi, \qquad\qquad \text{with probability } 1 - \eta \end{cases}, \tag{30}$$

where $\eta$ is anticipatory parameter. Store $\{s_t, a_t\}$ in $\mathcal{D}_{SL}$ if and only if the agent chooses an action based on $\varepsilon - \text{greedy}(Q)$.

NFSP also utilizes two innovations to ensure that the resulting algorithm is stable and can be simultaneously self-play learning [36]. First, it uses reservoir sampling [37] to avoid the window effect caused by sampling in a finite memory buffer. Second, it uses anticipatory dynamics [38] to enable each agent to sample its own best response behavior and more effectively track changes in the opponent's behavior.

NFSP uses the value-based DQN algorithm to solve the best response strategy. The double-DQN method solves the overestimation problem by separating the selection of the target action and the calculation of the target $Q$ value, and it can find better strategies [39]. The DRL algorithm combined with the dueling network architecture has a dramatic performance improvement [40]. Therefore, this paper adopted the double-DQN method combined with the dueling architecture to solve the best response. The loss functions for updating the parameters of the value network and the supervised network are given in (31) and (32), respectively [36,39].

$$\mathcal{L}\left(\theta^Q\right) = \mathbb{E}_{\{s,a,r,s'\}\in\mathcal{D}_{RL}}\left\{\left[r + Q\left(s', \arg\max_{a'} Q\left(s',a'\middle|\theta^Q\right)\middle|\theta^{Q'}\right) - Q\left(s,a\middle|\theta^Q\right)\right]^2\right\} \tag{31}$$

$$\mathcal{L}\left(\theta^\Pi\right) = \mathbb{E}_{\{s,a\}\in\mathcal{D}_{SL}}\left[-\log\Pi\left(s,a\middle|\theta^\Pi\right)\right] \tag{32}$$

*4.3. The Complete Game Process between the Radar and Jammer*

In the previous section, the confrontation relationship between the radar and the jammer was modeled through extensive-form games, and the NFSP method was given to solve the NE of the game. This subsection presents the complete competition process of the multiple-round interaction between the radar and the jammer, as shown in Algorithm 1.

---

**Algorithm 1** The complete game process between the radar and jammer.

---

1: Determine the radar action space according to the number of carrier frequencies and the number of subpulses
2: Determine the jammer action space based on the quantization step of jamming power
3: **for** each CPI **do**
4:　　Set the policy represented by (30) according to $\eta$
5:　　The radar observes the initial information state
6:　　**for** each pulse **do**
7:　　　　The radar chooses the transmission waveform as an NFSP agent
8:　　　　The jammer determines the number of look-through subpulses and the power allocation scheme as an NFSP agent
9:　　　　The radar receives an echo containing jamming signals
10:　　　　Calculate the SINR payoff according to (14)
11:　　　　The radar and jammer update their respective information states based on observed adversary behavior
12:　　　　Store transition experience $\{s_t, a_t, r_{t+1}, s_{t+1}\}$ in their respective $\mathcal{D}_{RL}$
13:　　　　**if** radar or jammer action $a_t$ is obtained by $\varepsilon - \text{greedy}(Q)$ **then**
14:　　　　　　Store $\{s_t, a_t\}$ in their respective $\mathcal{D}_{SL}$
15:　　　　**end if**
16:　　　　**for** each subpulse **do**
17:　　　　　　Judge whether to retain subpulses and update thresholds based on SAGC
18:　　　　**end for**
19:　　**end for**
20:　　Calculate detection probability based on SWD
21:　　Update network parameters $\theta^Q$ by (31)
22:　　Update network parameters $\theta^\Pi$ by (32)
23:　　Update target network parameters $\theta^{Q'}$ after a fixed number of iterations $\theta^{Q'} \leftarrow \theta^Q$
24: **end for**

---

## 5. Experiments

This section shows the competition results between the radar and the jammer under the jamming power dynamic allocation. The simulation experiments included detection probability training curves, a performance comparison between different quantization steps of jamming power, the verification of the approximate NE, the visualization of approximate NE strategies, etc. The basic simulation parameters are shown in Table 3.

**Table 3.** Basic simulation parameters.

| Parameters | Value |
|---|---|
| radar transmission power: $P_T$ | 30 kW |
| radar antenna gain: $G_R$ | 32 dB |
| radar initial carrier frequency: $f_0$ | 3 GHz |
| the number of pulses in one CPI: $N$ | 8 |
| the number of subpulses in one pulse: $K$ | 3 |
| the number of frequencies for the radar: $M$ | 3 |
| bandwidth of each subpulse: $B_n$ | 5 MHz |
| time width of each subpulse: $T_c$ | 10 μs |
| range between the radar and the jammer: $R$ | 100 km |
| false alarm rate | $10^{-4}$ |
| jammer transmission power: $P_J$ | 1 W |
| jammer antenna gain: $G_J$ | 0 dB |
| quantization step of jamming power: $P_{\min}$ | 0.2 |
| initial thresholds of SAGC | 0.5 |

According to Table 3, $M = 3, K = 3$, so the total number of actions of the radar is $\mathcal{A}_R = 27$. To decorrelate the subpulse echoes of different carrier frequencies, let the

frequency step size $\Delta f = 100$ MHz [17]. It was assumed that the RCS of the target does not fluctuate at the same frequency, but the RCS may be different at different frequencies [26]. Without loss of generality, the RCS corresponding to the three carrier frequencies was set to $[15, 3, 1]$ m$^2$. The number of samples of jamming power is five when $P_{\min} = 0.2$. Based on $P_{\min}$ and $M$, it can be known that there are 21 allocation schemes. Combining with $K$, then the total number of jammer actions is $\mathcal{A}_J = 64$. The radar actions and jammer actions are given in Figure 3.

| Action number | Action vector | | Action number | Action vector |
|:---:|:---:|:---:|:---:|:---:|
| 1 | [0, 0, 0] | | 1 | [1, 0, 0, 5] |
| ...... | ...... | | ...... | ...... |
| 7 | [0, 2, 0] | | 15 | [1, 2, 3, 0] |
| ...... | ...... | | ...... | ...... |
| 14 | [1, 1, 1] | | 30 | [2, 1, 2, 2] |
| ...... | ...... | | ...... | ...... |
| 20 | [2, 0, 1] | | 51 | [0, 1, 1, 3] |
| ...... | ...... | | ...... | ...... |
| 25 | [2, 2, 0] | | 57 | [0, 2, 2, 1] |
| ...... | ...... | | ...... | ...... |
| 27 | [2, 2, 2] | | 64 | [0, 5, 0, 0] |
| (a) radar actions | | | (b) jammer actions | |

**Figure 3.** The relationship between action number and action vector for the radar and the jammer.

As described in Section 4.2, this paper used the NFSP algorithm to train the radar and jammer. The NFSP algorithm contains a value network and a supervised network. Multilayer perceptron (MLP) [41] was used to parameterize these two networks in the experiments. The network information for DRL with the dueling architecture and SL is shown in Table 4 and Table 5, respectively.

**Table 4.** DRL network architecture.

| Layer | Input | Output | Activation Function |
|:---:|:---:|:---:|:---:|
| MLP1 | state size | 256 | LeakyReLU |
| MLP2 | 256 | 256 | LeakyReLU |
| MLP3 of Branch 1 | 256 | 128 | LeakyReLU |
| MLP4 of Branch 1 | 128 | 1 | / |
| MLP3 of Branch 2 | 256 | 128 | LeakyReLU |
| MLP4 of Branch 2 | 128 | action number | / |

**Table 5.** SL network architecture.

| Layer | Input | Output | Activation Function |
|:---:|:---:|:---:|:---:|
| MLP1 | state size | 256 | LeakyReLU |
| MLP2 | 256 | 256 | LeakyReLU |
| MLP3 | 256 | 128 | LeakyReLU |
| MLP4 | 128 | action number | Softmax |

The learning rates for DRL and SL were set to be 0.001 and 0.0001, respectively. The capacity for DRL memory $\mathcal{D}_{RL}$ and SL memory $\mathcal{D}_{SL}$ was 150,000 and 500,000. The update frequency of the target network parameters in the double-DQN was 4000. The anticipatory parameter $\eta$ of the mixed strategy was 0.1. The exploration rate of $\varepsilon - \text{greedy}(Q)$ was 0.06 at the beginning and gradually decayed to 0 with the increase of the number of episodes.

### 5.1. The Training Curve of Detection Probability

Let the game between the radar and the jammer go on for 400,000 episodes. Perform 1000 Monte Carlo adversarial experiments on the resulting policy every 2000 episodes to estimate the detection probability of the radar. The training curve is shown in Figure 4.

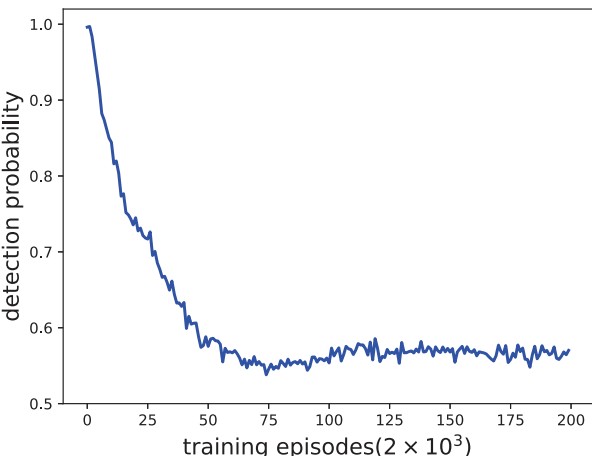

**Figure 4.** The detection probability curve.

It can be seen from Figure 4 that, as the number of training episodes increases, the detection probability gradually becomes stable and converges to 0.57.

In target detection theory, the detection probability is determined by the threshold and test statistic. If the statistical properties of the noise are known, the value of the threshold can be derived from the false alarm rate in constant false alarm rate (CFAR) detection. Then, the detection probability is determined by the test statistic. It can be known from the SWD algorithm that the SINR after coherent integration of each channel will affect the expression of the test statistic. Therefore, the results of the coherent integration directly affect the detection performance of the radar. Section 3.5 proposes to calculate the coherent integration of each frequency based on SAGC. It is clear from the calculation procedure of SAGC that the key to this criterion is the setting of the initial thresholds of the SINR. To illustrate the influence of the initial thresholds on the detection probability, five initial thresholds were set, as shown in Table 6. The radar and jammer strategies trained when $P_{\min} = 0.2$ were used to perform 1000 Monte Carlo experiments under different thresholds to obtain the variation of the detection probability with the thresholds. Figure 5 presents the result of this experiment.

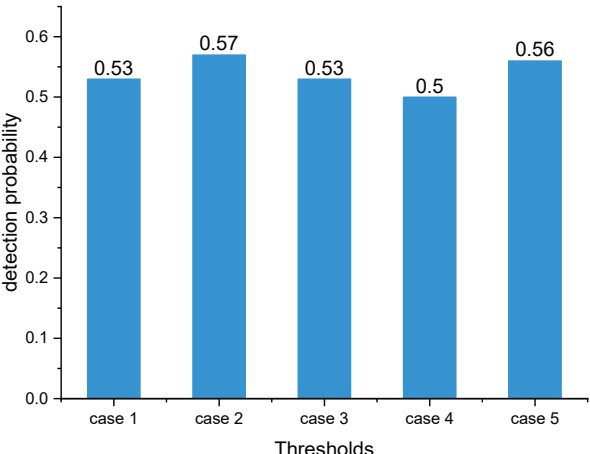

**Figure 5.** Detection probability of SAGC at different initial thresholds.

**Table 6.** Different initial thresholds.

| Thresholds | $T_0$ | $T_1$ | $T_2$ |
|:---:|:---:|:---:|:---:|
| Case 1 | 0 | 0 | 0 |
| Case 2 | 0.5 | 0.5 | 0.5 |
| Case 3 | 1 | 1 | 1 |
| Case 4 | 1 | 0.5 | 0 |
| Case 5 | 0 | 0.5 | 1 |

A coherent integration calculation method based on a fixed threshold criterion (FTC) was also adopted as a comparison. This method also needs to set thresholds. The calculation procedure is to retain the subpulse as long as the SINR is greater than the threshold. At the end of one CPI, the coherent integration for each frequency is calculated using the retained subpulses. Different from SAGC, the thresholds of FTC are unchanged in the whole training process, and the judgment of the current subpulse is only related to its SINR, not to the past retained subpulses. In contrast, the thresholds of SAGC are dynamic, and the judgment of the current subpulse needs to be combined with the past retained subpulses. Figure 6 shows the effect of different fixed thresholds (same as Table 6) on the detection probability under FTC. The experimental approach is to perform 1000 Monte Carlo with the radar and jammer strategies trained when $P_{\min} = 0.2$.

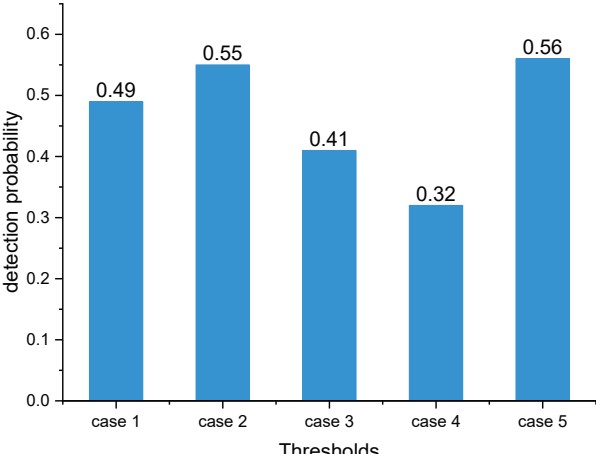

**Figure 6.** Detection probability of FTC in different fixed thresholds.

*Conclusion:* According to Figures 5 and 6, SAGC outperformed FTC. The reason for this result is whether to eliminate each subpulse depends not only on its SINR, but also on its contribution to coherent integration in SAGC. However, FTC only considers the subpulses themselves and does not care about the results of the coherent integration of all pulses.

*5.2. Performance Comparison between Different Quantization Steps of Jamming Power*

This subsection studies the performance comparison between different quantization steps of jamming power. Four quantization steps were set in the experiment: $P_{\min} = 1$, $P_{\min} = 0.5$, $P_{\min} = 0.2$, and $P_{\min} = 0.1$. The number of power samples in these four cases was 1, 2, 5, and 10, respectively. Therefore, the number of jammer action spaces corresponding to these situations was $\mathcal{A}_J^1 = 10$, $\mathcal{A}_J^{0.5} = 19$, $\mathcal{A}_J^{0.2} = 64$, and $\mathcal{A}_J^{0.1} = 199$. Figure 7 shows the jammer actions under different quantization steps.

| Action number | Action vector | | Action number | Action vector | | Action number | Action vector |
| --- | --- | --- | --- | --- | --- | --- | --- |
| 1 | [1, 0, 0, 1] | | 1 | [1, 0, 0, 2] | | 1 | [1, 0, 0, 10] |
| 2 | [1, 0, 1, 0] | | ...... | ...... | | ...... | ...... |
| 3 | [1, 1, 0, 0] | | 4 | [1, 1, 0, 1] | | 60 | [1, 7, 3, 0] |
| 4 | [2, 0, 0, 1] | | ...... | ...... | | ...... | ...... |
| 5 | [2, 0, 1, 0] | | 9 | [2, 0, 2, 0] | | 130 | [2, 9, 0, 1] |
| 6 | [2, 1, 0, 0] | | ...... | ...... | | ...... | ...... |
| 7 | [3, 0, 0, 0] | | 14 | [0, 0, 0, 2] | | 165 | [0, 3, 6, 1] |
| 8 | [0, 0, 0, 1] | | ...... | ...... | | ...... | ...... |
| 9 | [0, 0, 1, 0] | | 17 | [0, 1, 0, 1] | | 192 | [0, 7, 2, 1] |
| 10 | [0, 1, 0, 0] | | 18 | [0, 1, 1, 0] | | ...... | ...... |
| | | | 19 | [0, 2, 0, 0] | | 199 | [0, 10, 0, 0] |

(a) quantization step is 1 · (b) quantization step is 0.5 · (c) quantization step is 0.1

**Figure 7.** The relationship between action number and action vector for the jammer under different quantization steps.

The detection probability curves under different quantization steps are shown in Figure 8. From Figure 8, if the quantization step of jamming power is smaller, the detection performance of the radar is worse. However, the total number of jammer actions will increase accordingly, and the convergence speed will become slower. It can also be seen from Figure 8 that, when the quantization step is 0.1 and 0.2, the convergence results of the detection probability are consistent. This shows that the jamming effect of the jammer has performance boundaries.

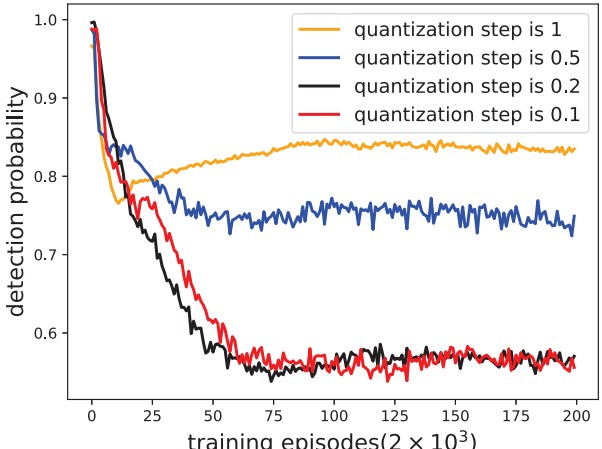

**Figure 8.** Detection probability curves of different quantization steps.

To verify whether the competition between the radar and the jammer can converge to an NE at the end of the training, the exploitability of the strategy profile needs to be evaluated. Exploitability is a metric that describes how close a strategy profile is to an NE [42–44]. A perfect NE is a strategy profile $(\sigma_1, \sigma_2)$ that satisfies the following conditions:

$$\begin{cases} u_1(\sigma_1, \sigma_2) \geq \max u_1(\sigma_1', \sigma_2) \\ u_2(\sigma_1, \sigma_2) \geq \max u_2(\sigma_1, \sigma_2') \end{cases}. \tag{33}$$

An approximate NE or $\epsilon$-NE is a strategy profile that satisfies the following conditions:

$$\begin{cases} u_1(\sigma_1, \sigma_2) + \epsilon \geq \max u_1(\sigma_1', \sigma_2) \\ u_2(\sigma_1, \sigma_2) + \epsilon \geq \max u_2(\sigma_1, \sigma_2') \end{cases}. \tag{34}$$

For a perfect NE, its exploitability is 0. The exploitability of $\epsilon$-NE is $\epsilon$. The closer the exploitability is to 0, the closer the strategy profile is to the NE. The exploitability curves under different quantization steps are shown in Figure 9.

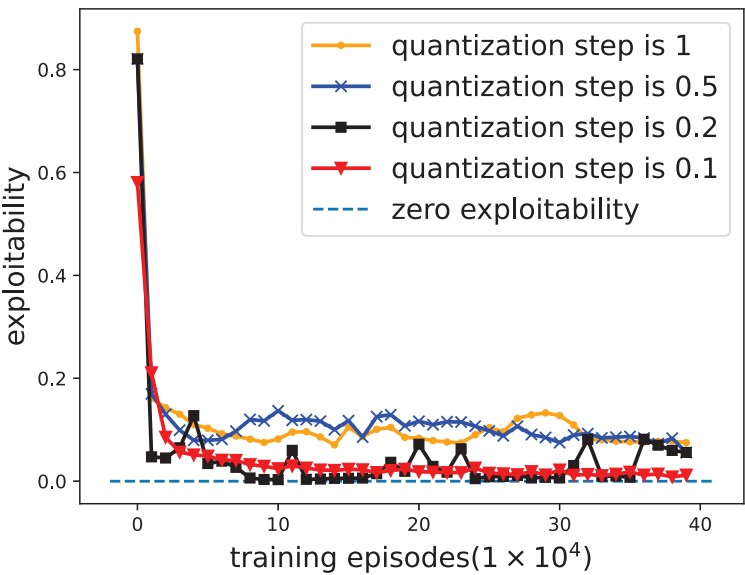

**Figure 9.** Exploitability curves of different quantization steps.

It can be seen from Figure 9 that, under different quantization steps, the exploitability curves gradually decrease and are close to 0. The exploitability when the quantization step is 0.1 and 0.2 can converge to 0. When the quantization step is 0.5 and 1, the exploitability converges to 0.05 and 0.07, respectively. This shows that the strategy profile of the radar and jammer can achieve an approximate NE under different quantization steps.

*Conclusion:* If the quantization step of jamming power is smaller, the total number of jammer actions will increase accordingly. Therefore, the jammer could explore the optimal jamming strategy so that the game between the radar and the jammer can achieve a real NE.

### 5.3. Visualization of Approximate Nash Equilibrium Strategies

Section 5.2 shows that the game between the radar and the jammer can converge to an approximate NE under different quantization steps of jamming power. Therefore, this subsection visualizes the approximate NE strategies. Through Figures 3 and 7, the corresponding relationship between the action number and action vector can be understood. The radar action vector is transformed into frequency, and the jammer action vector is transformed into power percentage for strategy research.

The strategies of the radar and jammer can be expressed in a three-dimensional coordinate system, in which the x-axis represents the action index, the y-axis represents the pulse index, and the z-axis represents the probability. Therefore, the meaning of the coordinates $(x, y, z)$ of any point is that the probability of choosing action $x$ at the $y$th pulse is $z$.

In Figures 10–13, (a) and (b) are the X-Y views of their strategies. The X-Y view shows the probability distribution of the radar or jammer's selection action on each pulse. (c) and (d) are the Y-Z views of their strategies. From the Y-Z view, it can be seen that the radar or jammer selects the action with the highest probability on each pulse.

In Figure 10, the radar prefers to select Actions 1 and 14, indicating that the carrier frequency combination of the transmitted signal is $[f_0, f_0, f_0]$ and $[f_1, f_1, f_1]$, respectively. The jammer tends to choose actions 165 and 192, representing that the power ratio allocated to $f_0$, $f_1$, and $f_2$ is $[0.3, 0.6, 0.1]$ and $[0.7, 0.2, 0.1]$. $\text{RCS}(f_0) > \text{RCS}(f_1) > \text{RCS}(f_2)$. The larger the RCS, the stronger the target echo power, so the jammer will allocate more power to reduce the SINR of the radar receiver. Jammer Action 192 allocates the most jamming

power to $f_0$, while there is little difference in jamming power between $f_1$ and $f_2$. Thus, the radar should choose $f_1$ with a larger RCS, corresponding to Radar Action 14. Jammer Action 165 allocates the most jamming power to $f_1$, so the radar selects $f_0$ with the largest RCS, corresponding to Radar Action 1.

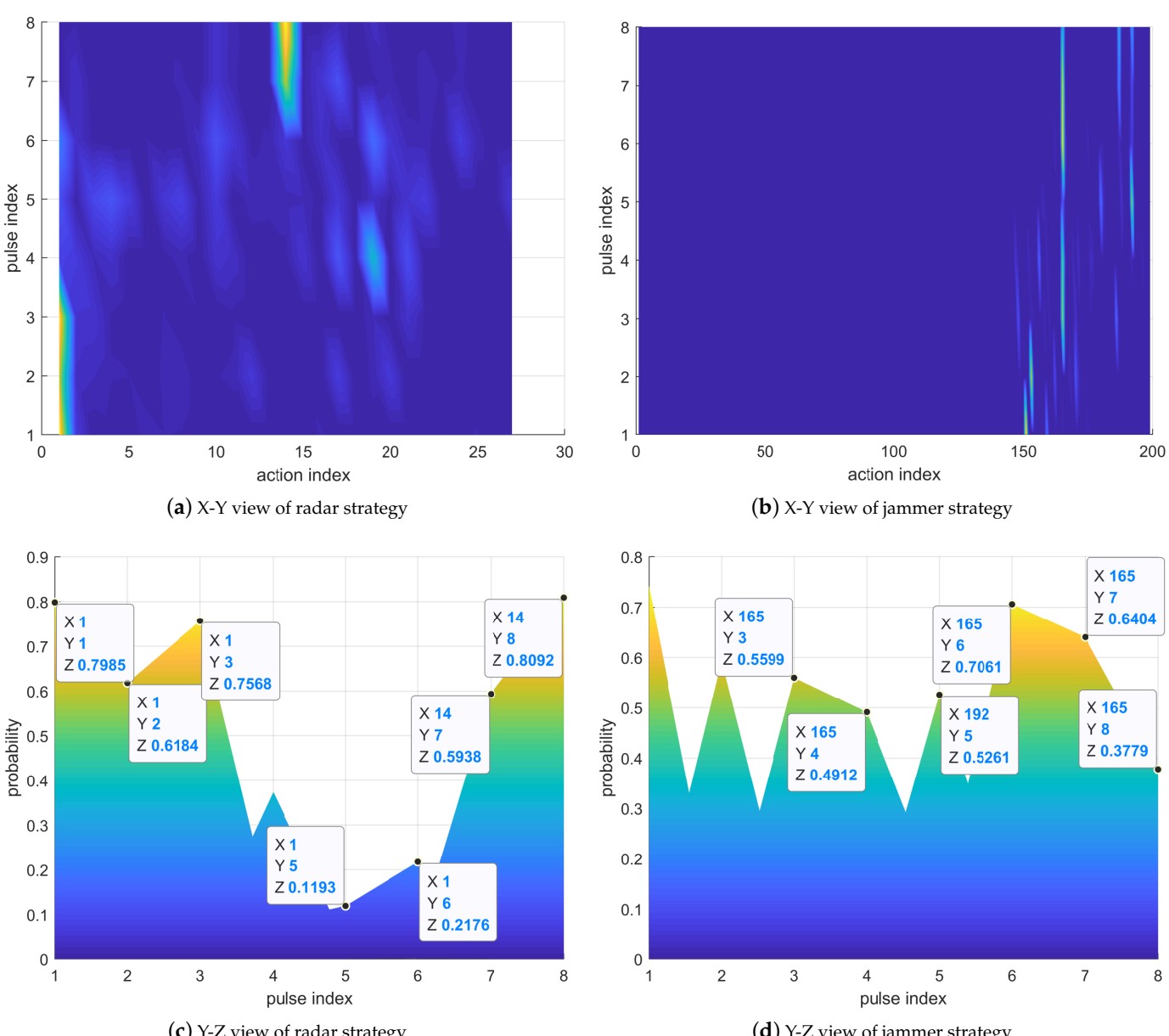

**Figure 10.** Approximate NE strategies with a quantization step of 0.1.

In Figure 11, the radar selects Action 1 with the highest probability. The jammer tends to select Action 57, indicating that the power allocated to the three frequencies is $[0.4, 0.4, 0.2]$. Although the power allocated by Jammer Action 57 to $f_2$ is the smallest, the RCS corresponding to $f_2$ is also the smallest, and the echo power is correspondingly the smallest. The jamming power of $f_0$ and $f_1$ is the same, but the RCS of $f_0$ is the largest. Therefore, the radar selects $[f_0, f_0, f_0]$, that is Action 1, which can ensure the maximum output SINR.

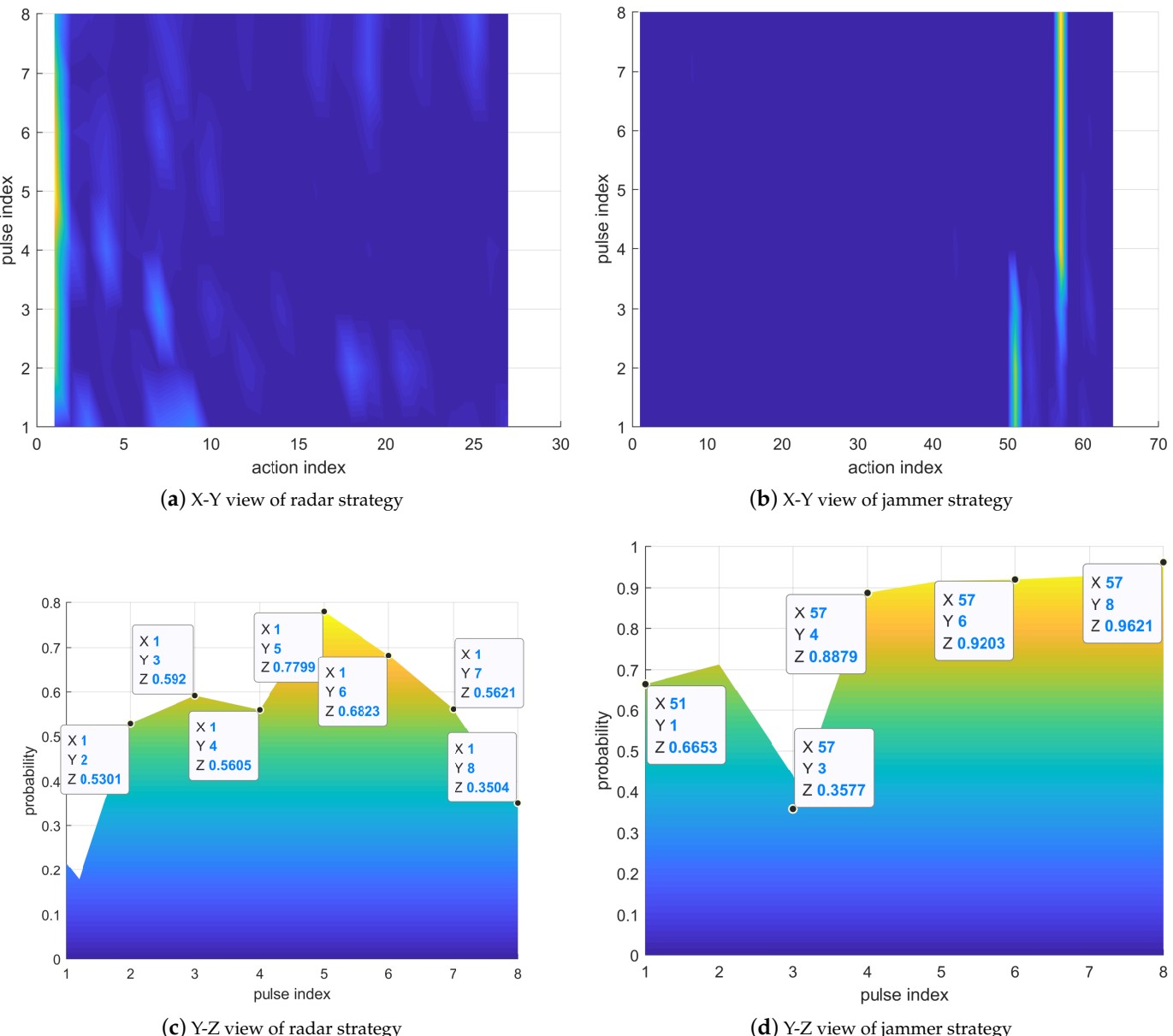

**Figure 11.** Approximate NE strategies with a quantization step of 0.2.

In Figure 12, the radar selects Action 27, meaning that the combination of the carrier frequency of the transmitted signal is $[f_2, f_2, f_2]$. The jammer selects Action 18, representing the power allocation scheme as $[0.5, 0.5, 0]$. The strategy of the jammer is to evenly distribute the power to the two frequencies with the first- and second-largest RCS. At this time, the radar selection Action 27 can ensure that all subpulses will not be jammed and the radar can obtain a larger SNR.

In Figure 13, the radar selects Action 14, which means the carrier frequency combination of the transmitted signal is $[f_1, f_1, f_1]$. The jammer selects Action 10, representing that the power allocation scheme is $[1, 0, 0]$, that is all the power is allocated to $f_0$ with the largest RCS. In this case, the quantization step of power is 1, so the jammer can only use all the jamming power to jam one frequency. At this time, the subcarrier of the subpulse of the radar is all $f_1$, which is the frequency of the second-largest RCS. In this way, it can not only avoid being jammed, but also ensure a large output SNR.

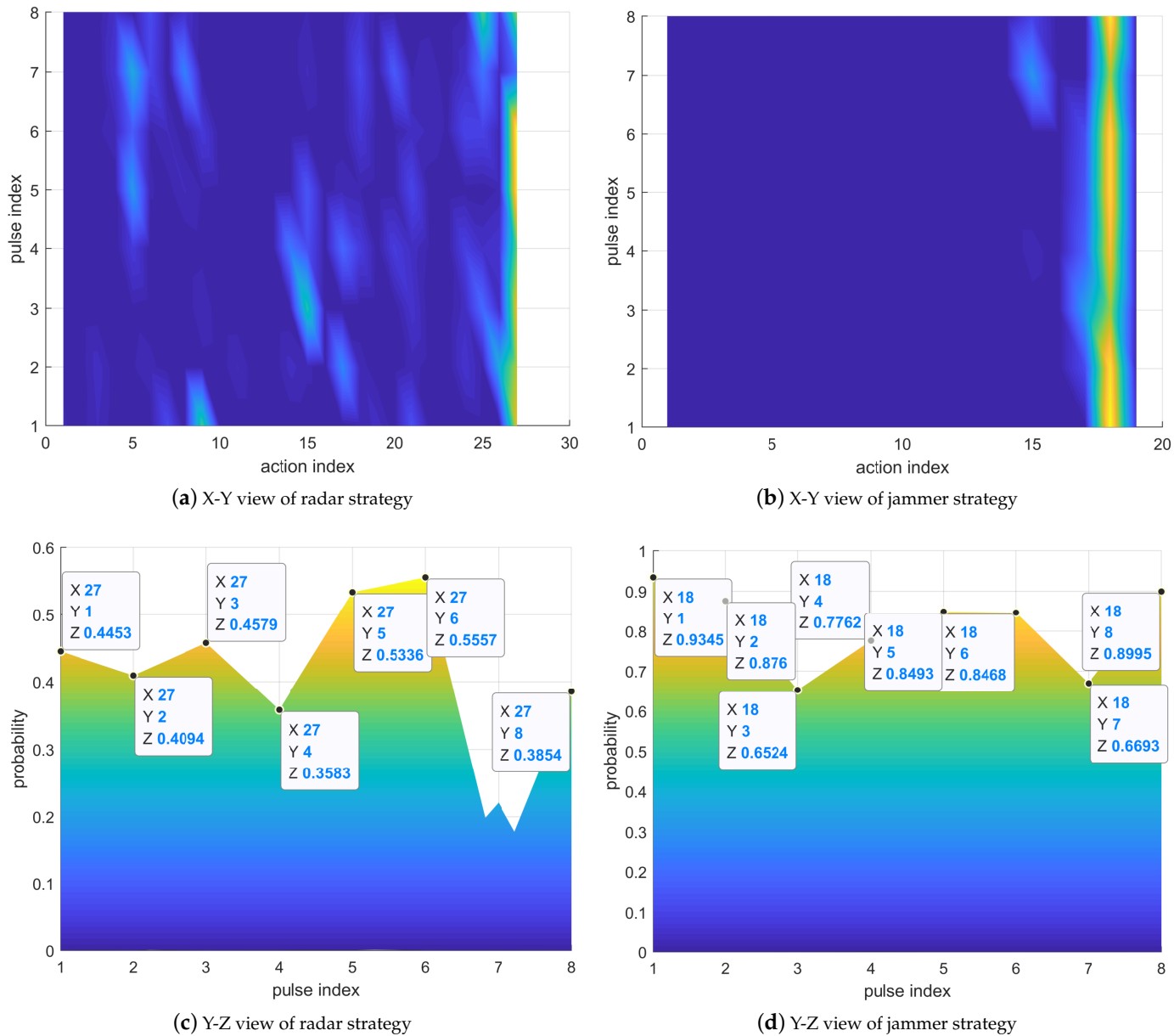

**Figure 12.** Approximate NE strategies with a quantization step of 0.5.

In these four different scenarios, when the game converges to the NE, the strategy of the jammer is that it does not perform the look-through operation. This shows that, when the jammer is regarded as an agent, it can learn the carrier frequency information of the radar through the interaction with the radar, so it only needs to optimize the power allocation strategy. In real electronic warfare, due to the limited confrontation time, the jammer cannot fully know the available frequencies of the radar, that is the jammer needs to intercept the subpulse of the radar most of the time, which indicates that the strategy of the jammer must deviate from the NE. Therefore, the radar can achieve better performance.

It can also be seen from Figures 10–13 that, no matter what the quantization step of jamming power is, the NE strategies of the radar and the jammer are mixed strategies. The radar and the jammer select actions from their respective action sets with a probability. This is the characteristic of imperfect information games.

*Conclusion:* Imperfect information games require stochastic strategies to achieve optimal performance [36].

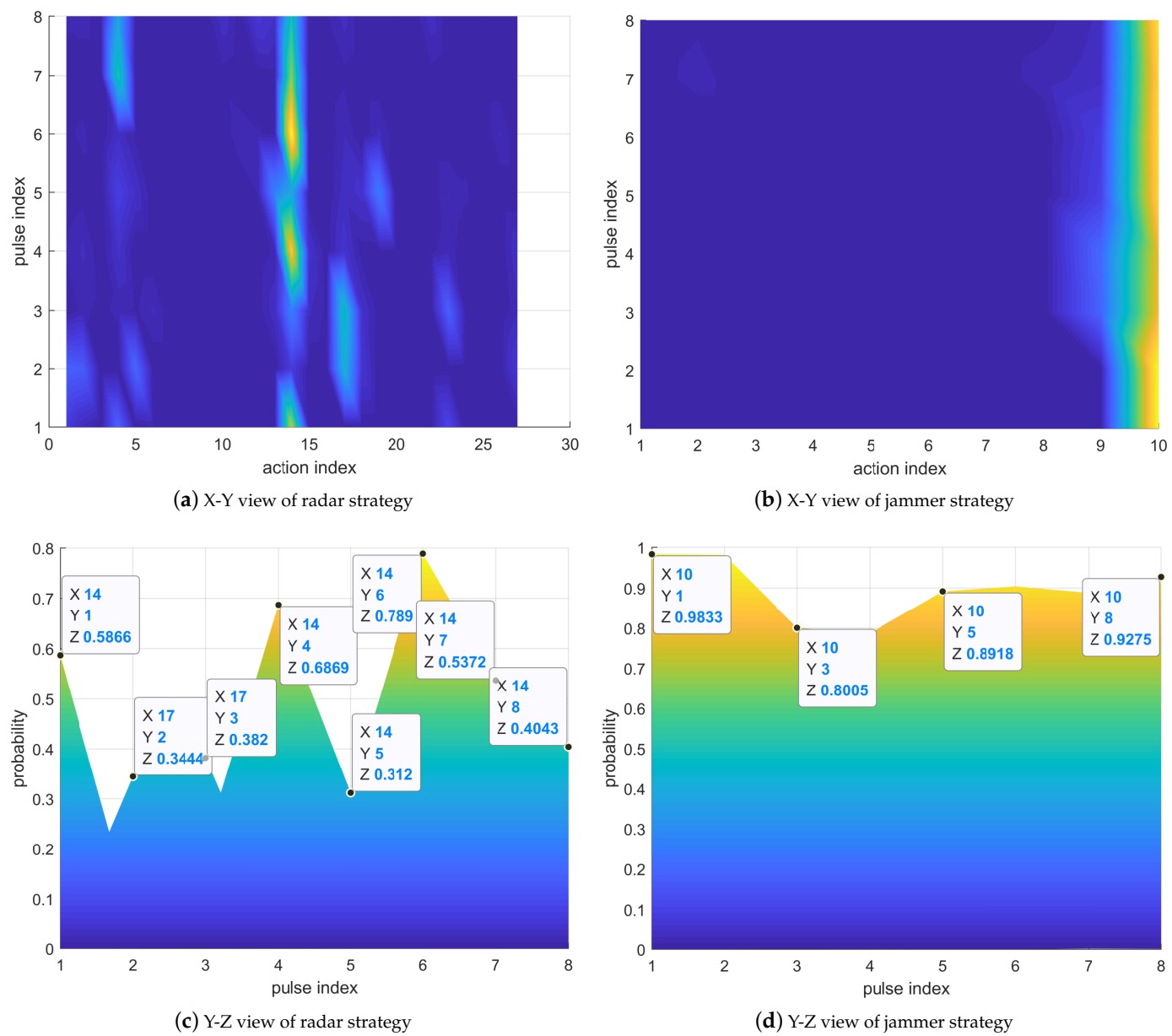

**Figure 13.** Approximate NE strategies with a quantization step of 1.

### 5.4. Comparison to Elementary Strategies

This subsection verifies the performance of the approximate NE strategies (ANESs) by comparing them with the elementary strategies.

Assume that the radar can choose two elementary strategies, which are the constant strategy (CS) and the stepped frequency strategy (SFS). The CS means that the carrier frequency of the radar is unchanged. Since the radar has three available frequencies, the CS includes three cases, denoted as CS0, CS1, and CS2. The SFS means that the carrier frequency of the radar increases or decreases step by step between pulses, and these two situations are recorded as SFS-up and SFS-down.

Two elementary strategies for the jammer were considered, which are the constant strategy (CS) and the swept strategy (SS). The CS means that the central frequency of the jamming signal remains unchanged. Similar to the CS of the radar, the CS of the jammer is also denoted as CS0, CS1, and CS2. The SS is similar to the SFS of the radar, and these two situations are recorded as SS-up and SS-down.

We made one side of the radar and jammer adopt the ANES, and the other side adopts the elementary strategies. In addition to the elementary strategies, the radar and the jammer also adopt the ANES as a comparison. The results of 1000 Monte Carlo experiments are shown in Figures 14–17.

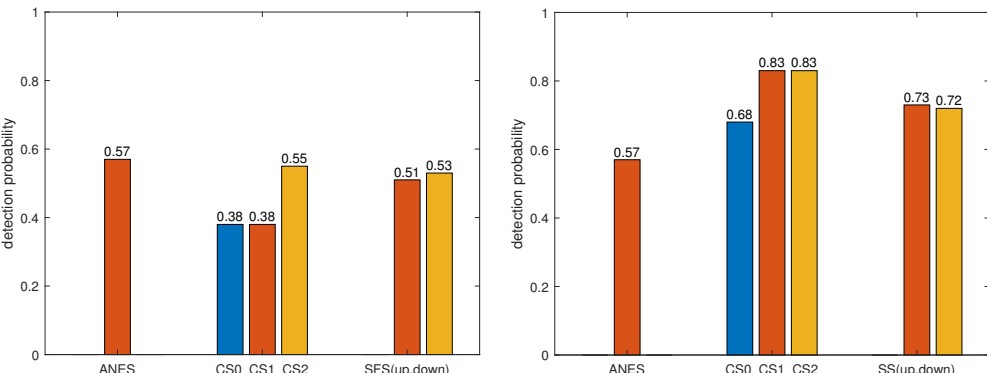

(**a**) The radar with elementary strategies and the jammer with ANES

(**b**) The radar with ANES and the jammer with elementary strategies

**Figure 14.** The quantization step of jamming power is 0.1.

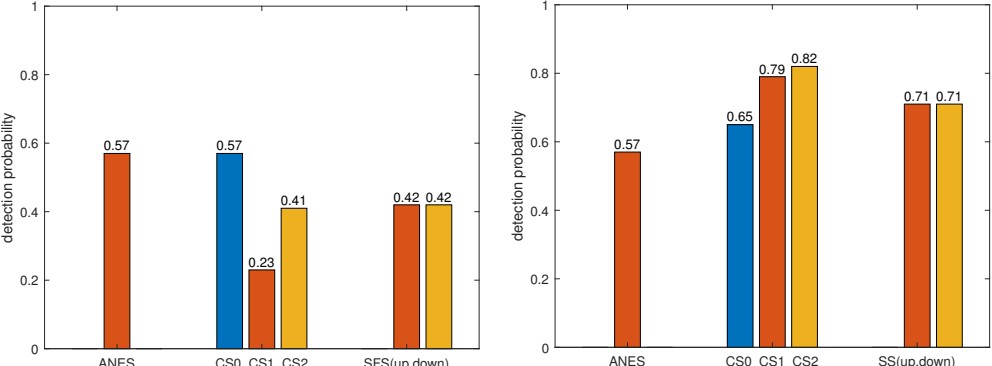

(**a**) The radar with elementary strategies and the jammer with ANES

(**b**) The radar with ANES and the jammer with elementary strategies

**Figure 15.** The quantization step of jamming power is 0.2.

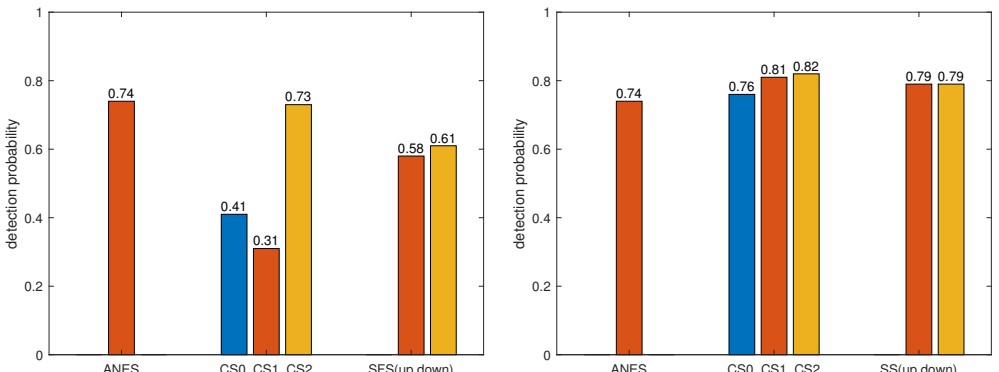

(**a**) The radar with elementary strategies and the jammer with ANES

(**b**) The radar with ANES and the jammer with elementary strategies

**Figure 16.** The quantization step of jamming power is 0.5.

In Figure 15, the detection probability of the radar adopting CS0 and the ANES is the same because these two strategies are similar in this jamming situation.

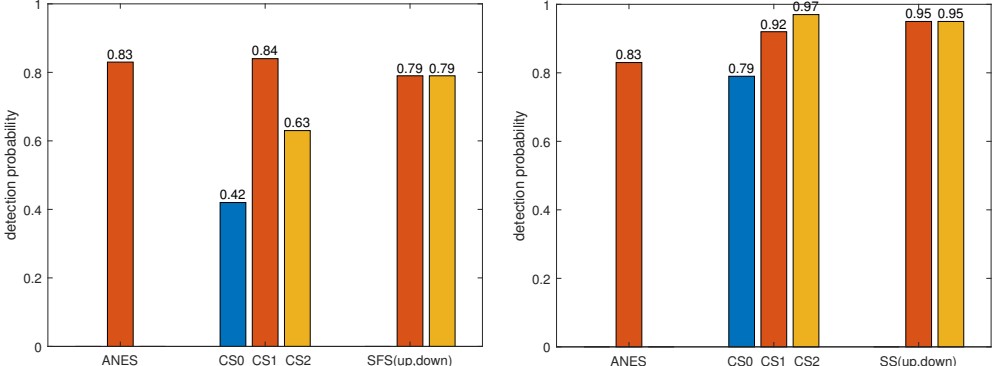

(**a**) The radar with elementary strategies and the jam-mer with ANES

(**b**) The radar with ANES and the jammer with ele-mentary strategies

**Figure 17.** The quantization step of jamming power is 1.

Similarly, in Figure 16, since the CS2 and ANES of the radar are the same, there is little difference in their detection performance.

In Figure 17, the ANES of the radar is the same as CS1, and the ANES of the jammer is the same as CS0. Therefore, the performance of one side adopting the ANES and the other taking the elementary strategy is basically the same as that of both adopting the ANES.

From Figures 14–17, the practical implication of the NE can be known, that is, as long as one side deviates from the NE, its performance will decrease. For the jammer, performance degradation refers to an increase in the detection probability of the radar.

*Conclusion:* The approximate NE strategies obtained in this paper are better than the elementary strategies from the perspective of detection probability.

### 5.5. Comparison to DQN

This subsection discusses the performance of the DQN in multi-agent imperfect information games. Two forms of the DQN were considered: DQN greedy and DQN average. DQN greedy chooses the action that maximizes the *Q* value in each state, so it learns a deterministic policy. DQN average draws on the idea of NFSP and also trains the historical average strategy through the supervised learning model, but the average strategy does not affect the agent's decision. Therefore, the agent chooses an action only based on $\varepsilon - \text{greedy}(Q)$ at each moment, not based on a mixed policy. DQN average can be achieved by setting the anticipatory parameter $\eta = 1$ in the NFSP algorithm. Because the NFSP agent in this paper solves the best response by the dueling double-DQN, DQN greedy and DQN average also adopt this method.

In Figure 18, the detection probability and exploitability curves of DQN greedy fluctuate markedly. Its exploitability cannot converge to 0, indicating that DQN greedy cannot achieve NE. Although the training curve of the detection probability of DQN average can be stable, its policy is highly exploitable. DQN average cannot reach an NE either.

*Conclusion:* DQN greedy learns a deterministic policy. Such strategies are insufficient to behave optimally in multi-agent domains with imperfect information. DQN average learns the best responses to the historical experience generated by other agents, but the experiences are generated only based on $\varepsilon - \text{greedy}$. These experiences are both highly correlated over time and highly focused on a narrow distribution of states [36]. Thus, the DQN average performs worse than NFSP.

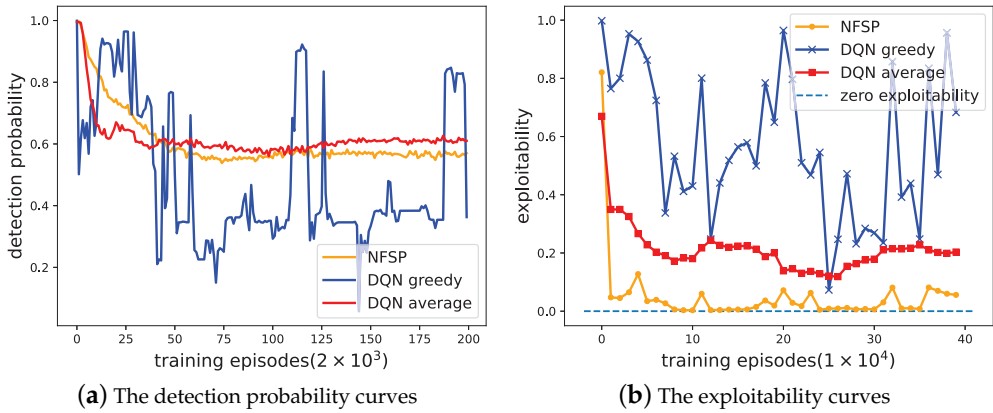

(**a**) The detection probability curves　　　　　　　(**b**) The exploitability curves

**Figure 18.** Comparison of three methods.

*5.6. Performance Comparison with Existing Methods*

To verify the effectiveness of the strategy obtained in this paper, a comparison between the proposed method and existing resource allocation methods was designed. The work in [17] is the strategy design problem based on RL, so the radar and the jammer interact with one of them as the agent and the other as the environment when applying this method to the established model of this paper. The strategy for the radar and jammer is solved independently rather than based on game theory. The work in [24] was based on the Stackelberg game and concluded that the jamming strategy is related to the target characteristic when the signal power is fixed. The method proposed in [25] was applied to the non-resource allocation scene, and the radar echo was processed by directly eliminating the jammed pulse. In addition to the above-mentioned methods, there is a common and without loss of generality method of allocating all power to the frequency with the second-largest RCS. This allocation strategy was proven by [25] to be feasible. This allocation method is denoted as a constant allocation strategy (CAS). The comparison result is given in Table 7.

**Table 7.** The comparison between the proposed method and other existing methods.

|  | **This Paper** | **Method in [17]** | **Method in [24]** | **Method in [25]** | **CAS** |
|---|---|---|---|---|---|
| detection probability | 0.57 | 0.61 | 0.65 | 0.61 | 0.79 |
| exploitability | 0 | 0.2 | 0.08 | 0.07 | 0.22 |

In Table 7, in addition to the proposed method in this paper, the exploitability of the other existing allocation methods cannot reach 0. Therefore, only the strategy obtained in this paper is an NE.

**6. Conclusions**

In this paper, the intelligent game between the subpulse-level FA radar and the self-protection jammer under the jamming power dynamic allocation was investigated. Specifically, the discrete allocation model of jamming power was established and the corresponding relationship between the quantization step of power and the available actions of the jammer was obtained. Furthermore, an extensive-form game model was used to describe the multiple-round sequence decision-making characteristics between the radar and jammer. A detection probability calculation method based on SAGC was proposed to evaluate the competition results. Then, due to the feature of the imperfect information game between the radar and jammer, we utilized NFSP, an end-to-end DRL method, to solve the NE of the game. Finally, simulations verified that the game between the radar and the jammer can converge to the approximate NE under the established model, and the approximate NE strategies are better than the elementary strategies from the perspective of

detection probability. The comparison of NFSP and the DQN demonstrated the advantages of NFSP in finding the NE of imperfect information games.

In the future, we should investigate the radar anti-jamming game with the continuous allocation of jamming power, in which the jammer has a continuous action space, and an algorithm to design the strategy for the radar and jammer should also be proposed.

**Author Contributions:** Conceptualization, J.G.; methodology, J.G. and K.L.; software, J.G.; validation, J.G., K.L. and Y.Z.; formal analysis, J.G.; investigation, B.J., H.L. (Hongwei Liu) and H.L. (Hailin Li); writing—original draft preparation, J.G.; writing—review and editing, B.J., K.L., Y.Z., H.L. (Hongwei Liu) and H.L. (Hailin Li); supervision, B.J.; funding acquisition, B.J., K.L. and H.L. (Hongwei Liu). All authors have read and agreed to the published version of the manuscript.

**Funding:** This work was supported in part by the National Natural Science Foundation of China under Grant 62201429 and 62192714, the Fund for Foreign Scholars in University Research and Teaching Programs (the 111 project) (No. B18039), the stabilization support of National Radar Signal Processing Laboratory under Grant KGJ202X0X, the Fundamental Research Funds for the Central Universities (QTZX22160).

**Data Availability Statement:** Not applicable.

**Acknowledgments:** The authors would like to thank all the Reviewers and Editors for their comments on this paper.

**Conflicts of Interest:** The authors declare no conflict of interest.

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
