# Peer review of "Radar and Jammer Intelligent Game under Jamming Power Dynamic Allocation"

_remotesensing, doi:10.3390/rs15030581_

Round 1
Reviewer 1 Report
I have some concerns which are listed below.
1) In subsection 3.2, the action of the jammer consists of interception and transmission, and you emphasize that no matter how many subpulses are intercepted, the number of the transmission action is the same. Therefore, whether the interception action is not needed?
2) In subsection 3.3, how do the information states reflect the imperfect information?
3) In subsection 3.5, you have proposed the SAGC method to calculate the detection probability. Why do you propose this method? In other words, what is the reason behind this approach?
4) In subsection 3.4, the symbols and represent the signal power and jamming power with respect to the th subpulse return, respectively. But in subsection 3.5, it seems that the symbols and represent the signal power and jamming power corresponding to the subpulse return with subcarrier . How to understand these two different definitions?
5) How to understand the meaning of formula (26)?
6) In Figure 4, what does the convergence of the training curve of the detection probability mean?
Reviewer 2 Report
The authors presented an interesting radar and jammer intelligent game under jamming power dynamic allocation strategy, in which the proposed zero-sum game converges to an approximate NE. The results of analyzes and simulations presented in this form can be successfully published. The reviewer is just asking you to review the content of the paper in terms of the so-called typos and text formatting.
1. In the field of radar resource optimization, power is generally treated as a continuous variable, but the author deliberately introduces "quantization step" to discrete the power variable in the paper. The author further explains the significance and advantages of quantization.
2. Please explain why the action set of the jammer [ 2,2,0 ] does not exist.
3. The authors need to supplement the undefined symbols in (14).
4. The authors are requested to streamline the content of the paper, especially to remove some repetitive expressions.
5. The simulation part only compared two forms of DQN. However, the author should also supplement the simulation part with the existing EC resource allocation method for comparison.
Reviewer 3 Report
The drafting of the proposed work is poor, especially avoid using "we" statements in the manuscript.
It is advised authors to add a comparison table of performance parameters between existing recent literature works and proposed work.
Round 2
Reviewer 2 Report
The authors have significantly improved the paper after this revision round. Before publication, the following comments should be carefully addressed.
Although it seems that investigating the quantization step of the power allocation has limited significance, the authors would have contributed to the resource optimization approach by introducing more of the DRL used in the manuscript.
